# Quantifying massively parallel microbial growth with spatially mediated interactions

**Florian Borse**[1], **Dovydas Kičiatovas**[1], **Teemu Kuosmanen**[1], **Mabel Vidal**[2], **Guillermo Cabrera-Vives**[2,3], **Johannes Cairns**[1], **Jonas Warringer**[4], **Ville Mustonen**[1,5]*

**1** Organismal and Evolutionary Biology Research Programme, Department of Computer Science, University of Helsinki, Helsinki, Finland, **2** Department of Computer Science, Universidad de Concepción, Concepción, Chile, **3** Data Science Unit, Universidad de Concepción, Concepción, Chile, **4** Department of Chemistry and Molecular Biology, University of Gothenburg, Göteborg, Sweden, **5** Institute of Biotechnology, University of Helsinki, Helsinki, Finland

* v.mustonen@helsinki.fi

**Data Availability Statement:** The codes used are available from GitHub: https://github.com/fborse/spatial-growth.

## Abstract

Quantitative understanding of microbial growth is an essential prerequisite for successful control of pathogens as well as various biotechnology applications. Even though the growth of cell populations has been extensively studied, microbial growth remains poorly characterised at the spatial level. Indeed, even isogenic populations growing at different locations on solid growth medium typically show significant location-dependent variability in growth. Here we show that this variability can be attributed to the initial physiological states of the populations, the interplay between populations interacting with their local environment and the diffusion of nutrients and energy sources coupling the environments. We further show how the causes of this variability change throughout the growth of a population. We use a dual approach, first applying machine learning regression models to discover that location dominates growth variability at specific times, and, in parallel, developing explicit population growth models to describe this spatial effect. In particular, treating nutrient and energy source concentration as a latent variable allows us to develop a mechanistic resource consumer model that captures growth variability across the shared environment. As a consequence, we are able to determine intrinsic growth parameters for each local population, removing confounders common to location-dependent variability in growth. Importantly, our explicit low-parametric model for the environment paves the way for massively parallel experimentation with configurable spatial niches for testing specific eco-evolutionary hypotheses.

## Author summary

Image-based platforms allow obtaining population size estimates for massively parallel growth experiments on substrate plates at a relatively low cost. However, such population size data has been shown to display a high degree of spatial variability, which occurs even with isogenic populations. Here we first quantified the importance of spatial location on growth variation using a machine learning approach, and then developed spatially aware

**Funding:** This work was supported in part by the Research Council of Finland (grant numbers 345829, 339496, 346128 to VM). https://www.aka.fi/ The funders had no role in study design, data collection and analysis, decision to publish, or preparation of the manuscript.

**Competing interests:** The authors have declared that no competing interests exist.

population growth models to explain the spatial structure of the growth data. Ultimately, we show that a spatial resource consumer model with local microhabitats connected via diffusion, and a parameter capturing the initial physiological state of a population, can fully explain the observed spatial variation in growth while allowing the inference of intrinsic growth parameters of specific populations. This result provides a method for systematic extraction of spatial growth models and paves the way for massively parallel eco-evolutionary experimentation.

## Introduction

Quantifying the growth of a population is a ubiquitous task in biology. Fields ranging from molecular life sciences and genetics to ecology and evolution require the ability to compare the growth of populations. Various methods have been developed to compare genetically similar populations subjected to different environments, as well as dissimilar populations subjected to the same environment—examples include frameworks such as quantitative fitness analysis [1] and synthetic genetic arrays [2].

Accurate quantification of growth is also critical for many real-life applications, such as assessing how fast a medical treatment will act on a pathogen population [3, 4]. Indeed, quantitative understanding of growth and how it can be modified is central to the emerging field of eco-evolutionary control [5].

There is a large variety of methods for cell cultivation, including growth in liquid [6] and solid [2, 7] medium. A range of population size quantification methods exist for cells grown in liquid medium, such as flow cytometry [8], although high accuracy usually comes at the cost of high financial expense and sophisticated equipment [9, 10].

In contrast, estimating the number of cells in a colony grown on solid medium relies on tackling different problems, such as the relationship between the three-dimensional structure of the colony [11, 12] and the diameter of the section in contact with the growth medium [13–15]. Nevertheless, the substantially cheaper and simpler equipment required for population size estimation can be used to justify the use of such methods despite reduced precision, as illustrated by numerous studies performed with such methods [16–18].

The advent of the high-throughput era has called for methods beyond sequencing that can be taken to a massively parallel level, stressing the utility of cost-effective and simple technologies. These include image-based platforms involving inexpensive cameras recording cell colonies growing on solid plates [10, 19–22]. In such platforms, a robot typically arranges and transfers colonies, and there is a method for obtaining population size estimates for each colony based on pictures taken automatically at regular time intervals.

This quantification process creates time series data which often involves several types of variability in measurements, often overlooked as noise or technical bias. One approach to handle such variability is to summarise the growth curves by reducing the curve to its most robust aspects, such as a single slope value or maximal growth rate [10], which is unfortunate considering the loss of information this operation entails. Indeed, understanding whole growth curves and not just a summary parameter can have important effects even to community co-existence and outcomes of evolution [23].

Moreover, when considering an experiment where populations grow together in a shared environment as a system of competing populations, treating the variation in population growth as mere noise seems to overlook ecological insights that could be obtained from such systems. Therefore, understanding that variability is an essential step not only for creating

procedures to reduce the variability but also for correctly interpreting the biological outcomes of the assays. For instance, it would be useful to ensure that the variability in genetically different populations originates from genetics instead of other sources of variability [2, 24].

Fig 1 shows the population growth time series of initially isogenic populations, each arranged on plate in a 32 x 48 grid that constitutes a shared environment, measured for four different environments (plates). Fig 1A plots the average of three aspects of the growth of a population $i$ as a function of time, namely: the population size estimates $N_i(t)$, their first derivative $\Delta N_i(t) = (N_i(t) - N_i(t - 1))/\Delta t$, and their relative growth rate $\rho_i(t) = \Delta N_i(t)/N_i(t)$. The latter rates are of particular interest, as they differ substantially from the single constant value one

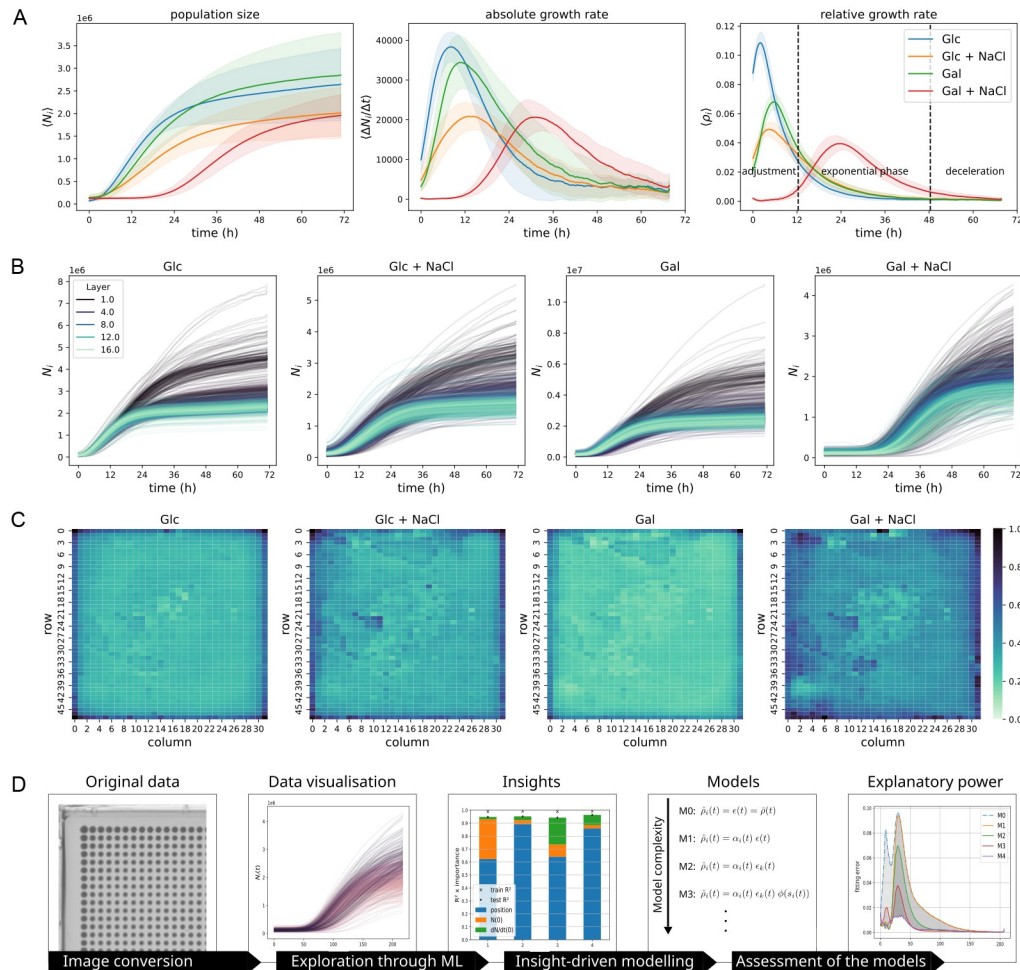

**Fig 1. Spatial patterns of population growth on semi-solid nutrient medium.** A total of 1536 isogenic populations were pinned in a 32 x 48 grid onto each of 4 plates, containing semi-solid nutrient media with small variations, and cultivated for 72 h with measurements of population size taken every 20 min. The nutrient media of the plates contain respectively 2% glucose only, 2% glucose + 1 M NaCl as growth-limiting substrate, 2% galactose only, and 2% galactose + 1 M NaCl (data from [10]). **A** Plate averages of population size estimates $N_i(t)$, absolute growth rates $\Delta N_i(t)$ and relative, per capita, growth rates $\rho_i(t)$, at each of the 218 time points. Relative growth rates offer an overview of the various growth phases, here annotated for the Gal + NaCl curve. **B** Population size $N_i(t)$ for all populations on each plate, coloured by layer of equivalent distances to the nearest grid border. Darker curves are closer to the border and exhibit greater growth. **C** Population size $N_i(t)$ at $t$ = 72 h for all populations on each plate (divided by the maximum value within each plate). Darker colours represent higher population size The four experiments were generated from the same pre-culture plate and will therefore share some growth features (see Methods). **D** Illustration summarising the workflow performed through this study.

would associate to exponential growth. For each plate there is a short such window, however, due to the variability of the start of growth of individual colonies, averaging blunts that region of constant growth which for individual colonies is also relatively short. The plate averages from Fig 1A mask a substantial amount of variability, visible in Fig 1B where all $N_i(t)$ population growth curves are plotted for each plate and grouped according to the distance in grid units to the nearest border of the grid. These curves clearly show a high level of variability—especially towards the later growth stages, where colonies located in the outer layer of the grid grow substantially and systematically more than their inner counterparts (Fig 1C) [10]. This effect is conserved between the four different growth environments, although it is most prominent for populations reaching the stationary phase of no net growth early. Because the stationary phase is reached when populations experience a nutrient and/or energy-source limitation, this suggests a disparity in nutrient and/or energy-source availability across these plates. Here, our main aim is to gradually build a mechanistic model, exploiting insight generated from non-mechanistic machine learning based analyses, to better understand these spatial effects.

Several previous studies have addressed the topic of competing populations in a shared environment (e.g., [25, 26]), and many growth models have been developed [27–29]. However, the notion of spatial distribution—especially for the shared environment—has been mostly formally addressed at the scale of a single growing colony [30–32]. A notable exception is the study by Chacón et al. on how colony size is being affected by neighbouring colonies due to the diffusion of limiting nutrients [33]. Their work highlights the biological relevance and importance of factoring spatial effects into studies of growth in a shared environment. Here we aim to describe a whole set of populations growing in a shared environment using a single model, allowing for a better understanding of the role of the spatial effects in interpopulation competition and extraction of intrinsic, strain-specific, growth parameters from such experiments.

To achieve this, we adopt a dual approach using both machine learning and mechanistic modelling and their interplay to gain biological insight (Fig 1D). Both approaches are needed as many important scientific problems of our time are not only about prediction at which machine learning excels. For instance, systematic control of the growth of microbial populations, with applications in antimicrobial therapy and biotechnology, needs a mechanistic basis for optimising over the space of possible control protocols that each will lead to some growth behaviour.

First, we quantify the importance of the location of a population for its growth dynamics using machine learning (ML) models. Machine learning algorithms, such as tree-based algorithms [34], tend to perform extremely well on prediction tasks. This allows us to determine the statistical importance of location when treated as an input feature and, at the same time, to obtain a general sense of how well population growth can be modelled using a set of chosen features. That is, narrowing down the set of features (parameters) which we should incorporate into a mechanistic model, such as initial population size or location.

Second, as the obtained ML models represent black boxes—i.e., their learned logic for prediction is not easily interpretable—we use the insights obtained from those models to incrementally develop mechanistically explicit models to capture and understand the effect of location.

Ultimately, we arrive at a mechanistic, low-parametric model which utilises nutrient and/or energy resource (henceforth resource) concentration as a latent variable and, through this variable, captures the interaction between populations and their local environment, as well as the diffusion of these resources across the global spatial level.

Identifying the stages at which our models substantially differ from the data allows us to determine the key components of growth.

## Results

### Machine learning models quantify determinants of growth and their relative importances

We wanted to quantify how much the spatial location, evident in Fig 1, and instantaneous population size, a key element in most growth models, contribute to population growth overall. Machine learning provides an excellent set of tools for this purpose, thanks to its ability to handle multivariate nonlinear relationships [35], which allows reducing these problems into individual regression tasks.

**Population size and location both affect growth rate and random forest regression captures the effect.** We first used only the instantaneous population size as an input feature, i.e., formally, $N_i(t) \mapsto \rho_i(t) \; \forall \; i, t$ where the model maps, for each population $i$, its population sizes at all points $N_i(t)$ to its relative growth rates $\rho_i(t)$. This results in low reconstruction errors across the four plates (see S1 Table). Adding location, $x_i$, of a population as an input feature, substantially increased train and test model performances (see S1 Table). Relative feature importance (see Methods) showed that population size and position contributed almost equally to model performance.

**Time-dependent random forest regressions reveal shifting importance of features predicting growth.** Next, we wanted to gain a more nuanced understanding of the relative importance of population size and location for the prediction of growth rate during the entire growth interval. We modified the previous location-aware regression task to independently map to $\rho_i(t)$ for each time $t$, resulting in a time-dependent series of regressions $N_i(t), x_i \mapsto \rho_i(t) \; \forall \; i$.

Fig 2A shows model performance as a function of time, using the $R^2$ measure to compare the model predictions $\hat{\rho}_i(t)$ with $\rho_i(t)$ computed from the measurements. Overall, a model that performed well was found for all plates, with only a few time intervals for which a substantial gap between the train and test data performances existed. The first low predictability window occurs during the lag phase (early part of growth), and is particularly visible for the slower-growing populations on the salt-containing plates. The second low predictability phase, present in each plate, occurs towards the end of the experimental series in which the performances decay, especially for the test data. Both windows coincide with the phases where net population growth is very small or non-existent, thus reflecting likely a measurement sensitivity threshold; growth rates below the threshold will be dominated by noise. Fig 2A additionally displays, for each regression, the importances of their input features, relative to the test data $R^2$ (see Methods). Several modes of growth can be observed: population position dominates the initial growth stages but becomes replaced by population size as the growth shifts to the exponential phase. Then, at a later stage, location predominates again as population growth rate decelerates and cells exit the exponential growth phase, while population size increases in importance as net growth plateaus. The phase of negative acceleration of growth [36]—also known as the deceleration phase [37]—is often associated with a progressive decrease in nutrient or energy availability. Thus, it seems reasonable to expect nutrient, or energy, availability to be an important factor in deciding the shifting importance of population location and size in models of growth across a plate.

**Summary variables of growth can be predicted by random forest regression and have variable main features.** To understand the interplay between location $x_i$ and common population growth model parameters—initial population size $N_i(0)$ and initial population growth $\Delta N_i(0)$—we devised three independent regression tasks, mapping these parameters to key summary values of growth: final population size $N_i(t_{\text{final}})$ (yield), maximum growth rate $\rho_{\max}$ (effective rate measured for each local colony directly from the data) and the timing of the latter $t_{\rho_{\max}}$.

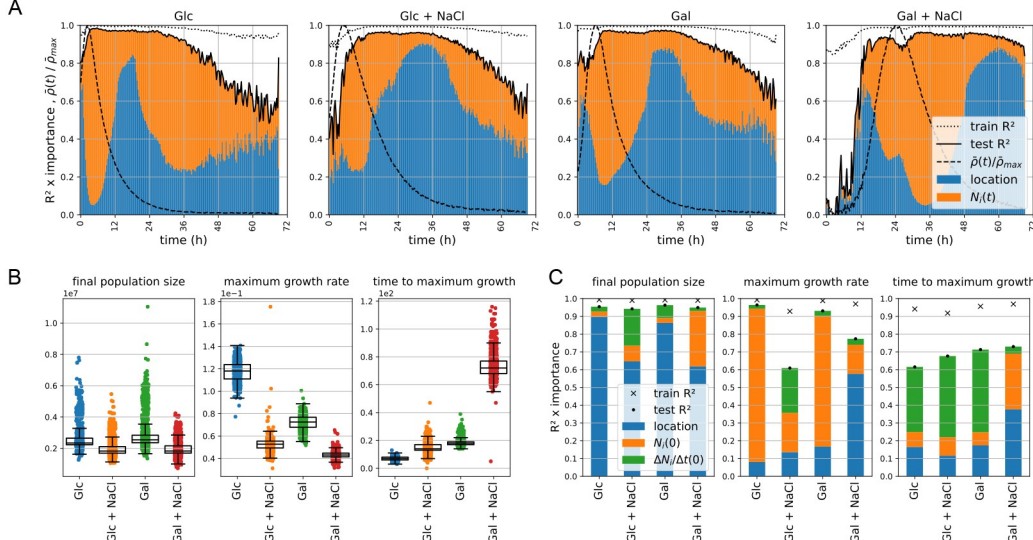

**Fig 2. Random forest regression reveals key determinants of growth and their relative importance across time. A** We trained random forest regressors to predict the relative growth rate $\rho_i(t)$ for every time point using two features as input: population grid position on the plate $x_i$ and population size $N_i(t)$. We evaluated the effect of these two features on model performance, in both train and test data sets, by multiplying feature importance by $R^2$ (see Methods). For long time periods of each of the four growth experiments (plates), we obtained high prediction scores for both the train and test sets. The short time periods with substantially lower model performance on test than on train data sets are discussed in the main text. Each panel represents the train and test scores for the trained models, and a contrasting view of the respective importances of location and $N_i(t)$. For ease of interpretation, we additionally superposed a normalised view of the relative growth curves. **B** Population growth is often described through key summary values, which common growth models use. Here we show the values we compute for three of them, namely the final population size (yield), the maximum growth rate, and its timing. **C** We trained random forest regressors on a set of three factors, the location in the grid, the initial population size $N_i(0)$ and the initial population growth rate $\Delta N_i(0)$ to predict the key summary values shown in Fig 2B. Whereas location is the predominant feature for predicting yield across the plates, the other two summary metrics are influenced more evenly by the three features and test data predictions are weaker for these metrics.

Fig 2C shows that $N_i(t_{\text{final}})$ size is dominated by location regardless of the composition of the growth medium, with very good prediction performance and little overfitting, as the differences between train and test scores are small. In contrast, $\rho_{\max}$ and $t_{\rho_{\max}}$ display a much more diverse interplay between the input features. While one might expect $\rho_{\max}$ prediction and its timing prediction to be mostly determined by respectively initial population size and initial population growth measurement, these assumptions do not hold—at least not for the Gal + NaCl environment. Similarly, the gaps between train and test data predictions that are visible for both salt-containing plates for $\rho_{\max}$ and across the plates for $t_{\rho_{\max}}$, are interesting.

## Explicit "mechanism-free" regression with location and population size specific components effectively capture growth variability

We showed above how the importances of position and population size vary across time when predicting population growth. During the exponential growth phase, population size has the strongest influence on growth, while at a later stage, position becomes the predominant determinant. Here we devise a set of models to explicitly understand these temporal and spatial effects.

**A location-agnostic null model reveals growth phases with high spatial and temporal variability.** Our baseline or null model, $\hat{\rho}_i(t) = \epsilon(t) = \bar{\rho}(t)$, consists of a single time-

dependent parameter $\epsilon(t)$ with no spatial awareness. We define our null model by comparing the growth rates to their average across a plate. Figs 3 and S1 display the fitting error as sum of squared errors (SSE) between its predictions of the growth rates $\hat{\rho}_i(t)$ and the rates $\rho_i(t)$ computed from the experimental data, labelled as the null model. The error rates are displayed both as spatial errors (Figs 3A, 3B and S1A–S1D) and temporal errors (Figs 3C, 3D and S1E). Clearly, a simple averaging of the growth rates does not capture spatial information of populations growing on a plate, and the spatial representations of the fitting errors display a clear structure where the outermost and innermost parts of the plates systematically show the most

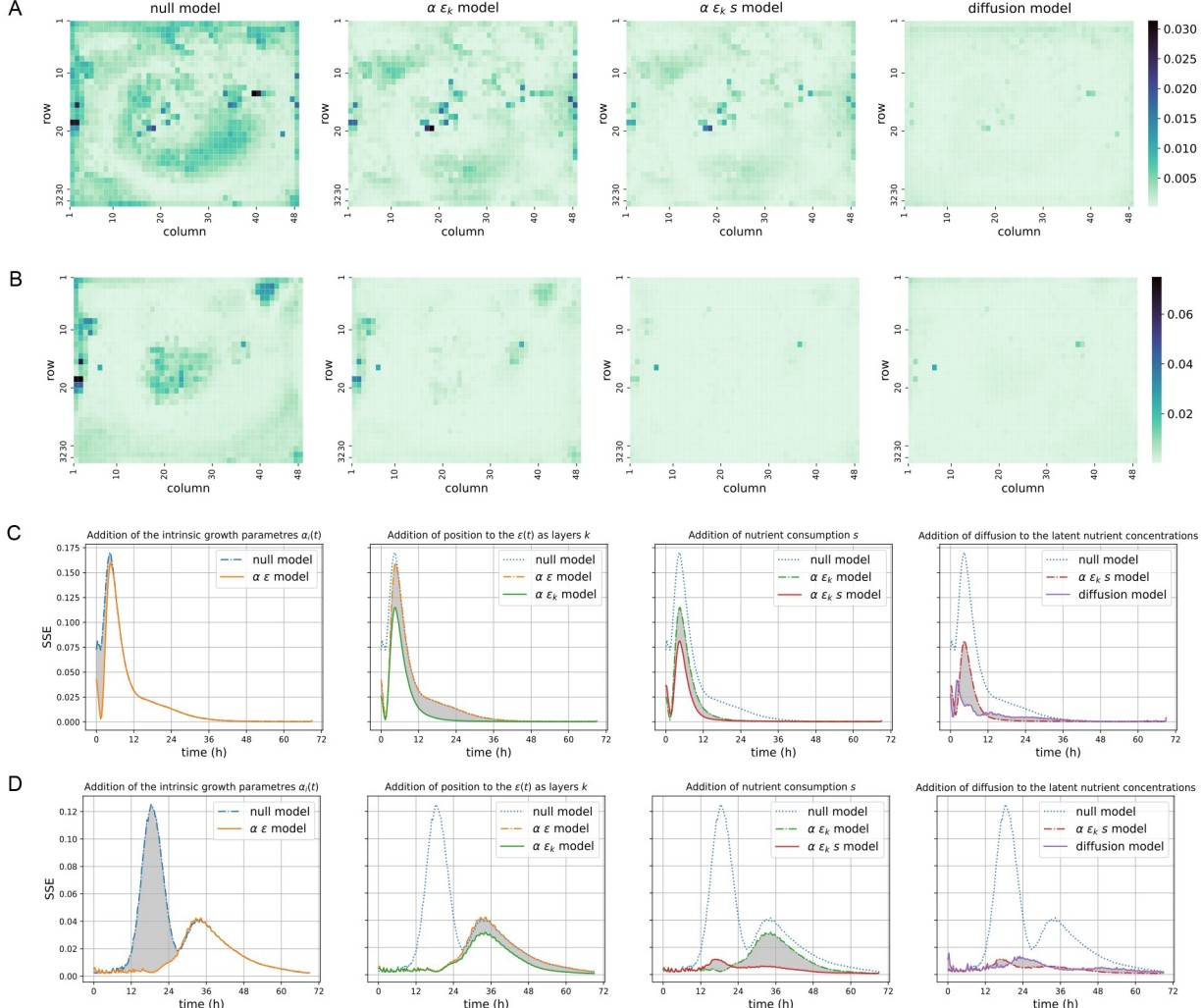

**Fig 3. Both temporal and spatial variability of growth within a plate can be captured by explicit growth models.** Panels represent, for each plate, the sum of squared errors (SSE) between the relative growth rates $\rho_i(t)$ from the experimental data and fitted rates $\hat{\rho}_i(t)$ from our models: 1) the null model, which simply averages the growth rates for all positions across a plate, 2) the population-specific $\alpha\epsilon$ model, where $\alpha_i(t)$ captures the physiological state of a population and the $\epsilon(t)$ is a mechanism-free parameter, 3) the population-and-location-specific $\alpha\epsilon_k$ model, where $\epsilon(t)$ is changed to $\epsilon_k(t)$ with $k$ indexing spatial locations, 4) the density-dependent $\alpha\epsilon_k s$ model, which adds a nutrient availability term $s$ to the earlier model, and 5) the diffusion model, a low parametric model, which further removes the mechanism-free parameter $\epsilon_k(t)$ and represents its effect through a diffusion process. **A—B** The spatial representation of the fitting errors for two environments—Glc and Gal + NaCl —where the SSE are computed for each population individually across all the time points. **C—D** The temporal representation of the fitting errors for two environments—Glc and Gal + NaCl—where the SSE are computed for all the populations in either of the two plates at every time point. Each panel represents an additional comparison from one model to the previous by grey areas highlighting the differences between the model errors. Both density dependent and diffusion model almost fully remove spatial and temporal variability.

error. Interestingly, the temporal representations of the fitting errors display an analogous structure: two bumps of high errors, the first one coincidentally occurring during the exponential growth phase, which may be attributed to populations switching to growth at different rates. Therefore, such a colony identity agnostic model fails to account for these differences in start of growth, on top of failing to capture the spatial effect.

**Population-specific state parameter captures the variability during the exponential growth phase.**   Baranyi & Roberts [28] suggested modelling population growth curves by the growth term $\alpha_i(t)$, modulating the timing of the start of the exponential growth stage:

$$\alpha_i(t) = r_0 \frac{c_i}{c_i + e^{-mt}} \tag{1}$$

where $r_0$ represents the maximum growth rate, $c_i$ the internal physiological state of the cell population, and $m$ the rate at which the cells change their internal state. Such an adjustment function codes population-specific information through its $c_i$ parameter, while the other two parameters are assumed to be global, i.e., the same for every isogenic population. This is justified as in the initial phase of the growth experiments, populations do not compete for nutrients yet, hence their environment is sufficiently homogeneous to consider $r_0$ and $m$ global. Thus, $c_i$ allows for modelling the growth of a population by taking into account variability in the initial physiological states:

$$\hat{\rho}_i(t) = \alpha_i(t)\,\epsilon(t). \tag{2}$$

Figs 3 and S1 label this model as the $\alpha\epsilon$ model. Fig 3C and 3D show clearly that the the first bump in the temporal errors, corresponding to the exponential growth phase, is almost fully removed—meaning that the incorporation of $\alpha_i(t)$ captures the population-specific elements affecting initial growth across the plates—while the second bump is still present.

**Extending the mechanism-free parameter to spatial layers partially captures the growth deceleration phase.**   Fig 1B hints that population growth can be grouped along their distance to their closest grid border. In particular, we assume that the growth environments of the populations will differ similarly along such distance. We utilise this information to constrain a location-specific mechanism-free parameter $\epsilon(x_i, t)$ to $k$ layers of locations equidistant to their closest grid border, which aims at capturing the biases induced by variably growing populations in the later phases of an experiment:

$$\hat{\rho}_i(t) = \alpha_i(t)\,\epsilon_k(t). \tag{3}$$

Figs 3 and S1 label this model as the $\alpha\,\epsilon_k$ model. The difference between its fitting errors and those from its spatial-location-agnostic counterpart can be relatively subtle depending of the plate considered. Overall, the temporal representations of these errors display a moderate but significant reduction in error rates, except for the Glc + NaCl environment (S1 Table shows for successive models the improvement over the null model using the Akaike information criterion (AIC), where a lower score represents a better ability to fit while taking into account the number of parameters used).

**Adding a population size dependent resource consumption term greatly improves the model.**   Machine learning results presented above highlighted, in addition to location, a substantial importance of the local population size $N_i(t)$ in predicting growth. That local population size should affect growth is clear from basic considerations of the underlying birth–death type processes. In addition, the coupling of local environmental state and $N_i(t)$ seems expected. While the populations on the same plate are here isogenic and their local environment initially the same everywhere across a plate, differently growing populations affect their growth

environment differently, which creates a visible effect as an experiment runs. Incorporating $N_i(t)$ into a regression model could be accomplished in several ways. Here we adopt a minimalistic approach that fits our context. First, we denote the state of the local growth environment (resource concentration) by $s_i(t)$ and assume that the function converting resource concentration to growth (e.g., the standard Monod type of model [27]$s/(K+s)$) can be linearised (e.g. in Monod $K \gg s$). We can then write the relative growth rate as:

$$\hat{\rho}_i(t) = \alpha_i(t) \; \epsilon_k(t) \; s_i(t). \tag{4}$$

Next we model the environmental change caused by population growth at a given time as a linear process [38]:

$$\frac{ds_i(t)}{dt} = -v \frac{dN_i(t)}{dt} = -v\alpha_i(t) \; \epsilon_k(t) \; s_i(t)N_i(t) \tag{5}$$

Taking the observed $N_i(t)$ as given (i.e., a constraint), we can express the latent resource dynamics as

$$s_i(t) = s_i(0)e^{-\int_0^t v \, \alpha_i(t') \, \epsilon_k(t') \, N_i(t')dt'}, \tag{6}$$

which can be substituted back to Eq 4 leading to a population density dependent growth model with a single additional global parameter compared to the earlier model—the resource consumption rate $v$ (we set $s_i(0) = 1$, i.e., express the concentrations in arbitrary units). The special case where both $\alpha_i(t)$ and $\epsilon_k(t)$ would be constants corresponds to the standard logistic growth model.

The temporal representations of the errors (Fig 3C and 3D)—referred to as density-dependent—show for every plate a substantial improvement of the fits, mediated primarily by a significant improvement in the deceleration phase. This is particularly apparent for the Gal + NaCl environment. The spatial representations of the errors also show a close to zero error rate at the border locations in this same environment.

## Low-parametric resource consumer model with spatial diffusion of resources effectively captures growth variability

Here we show that a formulation of growth based on resource-consumer dynamics, which is exploiting the latent space of spatially coupled resource concentrations, results in a low-parametric regression model that effectively captures growth rate variability across time and space. Although the last model discussed, Eq 4, is explicit compared to the ML model and explains the data very well across the plates, it still has two major drawbacks. First, the spatial field $\epsilon_k(t)$ can be thought of as a black box itself, that provides no real insight into the phenomenon. Second, $\epsilon_k(t)$ uses 16 parameters for each time point and cannot be really considered as low-parametric in every use case. Although here, with isogenic populations learning $\epsilon_k(t)$, it is still sensible (1536 data points for each time point), for a genetically heterogenous plate design one would need to reflect on whether $\epsilon_k(t)$ would be genotype-dependent, possibly making the entire inference unfeasible. Here we reduce the model dimensionality of the problem by replacing these mechanism-free parameters with a clearly defined process, which allows for generalisation of the approach to multiple genotypes/strains in a plate.

**Explicit model of colony, environment and inter-environment interactions.** The populations grow on a medium where an agar polymer stabilises a slow-moving liquid, which carries the nutrients and energy sources the populations consume. This suggests that we can replace the mechanism-free spatial component with a global diffusion process $D\nabla^2 s(x_i, t)$,

which unites the local growth environments utilising a plate-wide approach. This allows for the location of a population to matter, both compared to its surrounding populations and the distance to the borders of the grid, thus further influencing the dynamics of its environment. Therefore, description of the latent variable, representing the available resources, assumes the following form:

$$\frac{\partial s(x_i, t)}{\partial t} = D \nabla^2 s(x_i, t) - F(N_i(t)) \tag{7}$$

where $F(N_i(t))$ is a term describing the interaction of a population $i$ with the environment located at $x_i$.

The model presented in the previous section assumed linear consumption of the resource, proportional to absolute growth rate. We further extend this term by incorporating another linear consumption term, referring to the maintenance of the population size; therefore, this yields a resource consumption term

$$F(N_i(t)) = v_1 \frac{dN_i(t)}{dt} + v_2 N_i(t) \tag{8}$$

which represents for our purposes the interaction between a population and its environment.

The previous model, Eq 4, assumed that the coupling function that maps the resource concentration to growth is simply linear in $s$. As we remove the dependency on the mechanism-free parameter $\epsilon_k(t)$, we allow a more general coupling function $f(s(x_i, t))$ (see Methods), which effectively parameterises a family of monotonic nonlinear functions. Thus, population growth can be written via the intrinsic growth term and the effect of local nutrient availability, which leads to the following equation:

$$\hat{\rho}_i(t) = \alpha_i(t) f(s(x_i, t)). \tag{9}$$

Importantly, both resource and population models, Eqs 7 and 9, become coupled across the whole plate. This coupling results in a more elaborate inference protocol that we solve by an iterative algorithm (see Methods). Figs 3 and S1 show the performance of the model (referred to as diffusion model) in detail. The spatial fitting errors across the plates have almost fully disappeared, meaning that the spatial information is—especially for the salt-containing plates—almost fully captured.

The temporal errors displayed in Figs 3 and S1 show significant improvements compared to the diffusion-agnostic models, which are especially impressive for the salt-free plates. In comparison, the model shows for the salt-containing glucose plate a somewhat less impressive improvement, and for its galactose counterpart the performances at the initial phase seem worse. The source of this phenomenon is difficult to assess from the spatial representation of the fitting errors, as only a few locations seem to add significant amounts of error. In any case, the diffusion model captures the growth data across all plates very effectively and represents a mechanistic and low-parametric model for the spatial variability that is evident in the observed data.

The inferred parameters (see Table 1) allow us to compare the growth characteristics of the studied yeast strain across the conditions without spatial bias. The intrinsic growth rate, $r_0$, is largest in glucose and adding salt reduces it approximately by 56%. Pure galactose growth rate is about 65% of that obtained in glucose and adding salt reduces it by about 50%. The rate of changing the physiological state of the populations, $m$ (inversely related to lag-time), is largest in the glucose condition with salt lowering the rate by $\sim 30\%$. Galactose has $\sim 45\%$ lower rate

**Table 1. Table of global inferred parameters of the diffusion model.**

|         | Glc | Glc + NaCl | Gal | Gal + NaCl | units |
|---------|-----|------------|-----|------------|-------|
| $r_0$   | 0.119 | 0.052 | 0.078 | 0.040 | $[20 \text{ min}^{-1}]$ |
| $m$     | 0.417 | 0.288 | 0.229 | 0.138 | $[20 \text{ min}^{-1}]$ |
| $D$     | $3.60 \times 10^{-6}$ | $4.33 \times 10^{-6}$ | $5.15 \times 10^{-6}$ | $3.79 \times 10^{-6}$ | $[\text{cm}^2/\text{s}]$ |
| $v_1$   | $1.16 \times 10^{-5}$ | $7.63 \times 10^{-6}$ | $5.56 \times 10^{-6}$ | $2.65 \times 10^{-6}$ | a.u. |
| $v_2$   | $7.05 \times 10^{-8}$ | $1.46 \times 10^{-7}$ | $4.09 \times 10^{-8}$ | $2.39 \times 10^{-8}$ | a.u. |
| $K$     | 16.781 | 16.599 | 6.158 | 4.774 | a.u. |
| $\kappa$ | 0.007 | 0.007 | 0.032 | 0.083 | none |

As the resource dynamics are latent we were not able to fix the absolute units for consumption and maintenance rates $v_1$ and $v_2$ without additional assumptions. The shape parameter $K$ would have a unit 1/[resource] which is expressed in the model in arbitrary units.

compared to glucose and adding salt reduces the rate by $\sim 40\%$. As expected, these considerations for $r_0$ and $m$ qualitatively match the data plotted in Fig 1A.

The diffusion rates are similar between all four conditions and range from about 50% to 70% to what has been reported for glucose in pure water [39]. We assessed the impact of diffusion to growth by studying the latent resource space (inferred as a part of the diffusion model), averaged across the layers of the plates and compared to the mean of the neighbouring local resource fields (Fig 4A). For each plate the three outermost layers show the biggest effect of diffusion, reflecting the surplus of resources coming from the colony free border areas of the plates. A rule of thumb estimation of the absolute importance of the diffusion effect across the growth conditions can be done by considering the product of $Dr_0v_1$ which takes into account the growth of a population, its impact to the local resource and the diffusion of resource across the plate. This measure results in an order of the effects: glucose > galactose $\sim$ glucose + salt > galactose + salt, closely matching what is visible in Fig 4A.

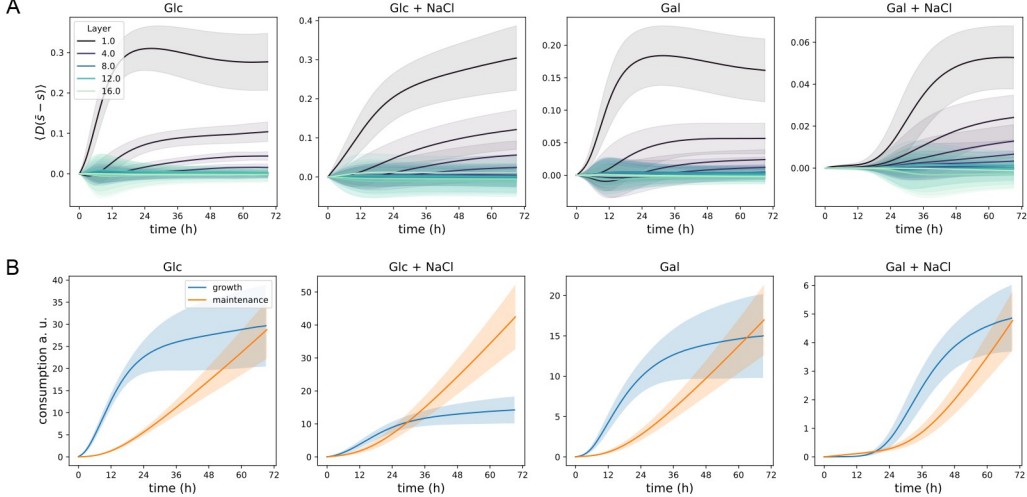

**Fig 4. Latent resource concentrations show the impact of diffusion on growth and allow to study resource usage. A** We assessed the impact of resource diffusion to growth by evaluating the mean of $D(\bar{s}_i - s_i)$ across plate layers from the diffusion model inference (shading shows the variance across time of the considered term). The effect of resource diffusion is substantial for colonies located at the three outermost layers. **B** We computed how much the populations would have consumed resources according our model Eq 8 by taking the time integral of growth and maintenance terms with the inferred rates and using the actual measured population dynamics. Resource usage for growth dominates initially, however, maintenance catches up for all but the glucose with salt condition.

The comparison of both consumption for growth, $v_1$, and maintenance, $v_2$, across conditions is more challenging as they remain in arbitrary units. This is because we do not observe the amount of the resource directly but it stays as a latent variable in the experiment. Although we know the initial media composition (see Methods) we do not have another measurement, e.g., the end point composition to fix an absolute scale. We can, however, compare first how a population would have consumed resources according to our model Eq 8 by taking the time integral of growth and maintenance terms with the inferred rates and using the actual measured values for $dN/dt$ and $N$ (see Fig 4B). For glucose we see that consumption for growth dominates initially but the total usage of resources at the end is about equally divided between growth and maintenance. Adding salt leaves the maintenance curve very similar but substantially lowers the amount of resource used for growth within the duration of the experiment. In galactose both with and without salt the usage of the resource is roughly equally divided between growth and maintenance at the end. Similarly to glucose, adding salt leads to smaller usage of the resource to growth—this is also visible from Fig 1A average yields.

Shape parameters $K$ and $\kappa$ of the function $f(s)$, which maps the resource concentration to growth, are in themselves somewhat hard to interpret. However, the biology encoded by these parameters is directly observable by plotting the functions. Fig 5 shows that the concentration to growth functions in galactose conditions decrease faster as resource concentration get lower than those in glucose.

Our diffusion model analysis allows to infer global intrinsic growth parameters as opposed to effective parameters that one would get by considering each population in isolation and absorbing, inadvertently, variations in microenvironments to effective growth parameters. In particular, the global model of the shared environment allows us to further probe the final local parameter of our mechanistic model, the physiological state $c_i$. We first tested the flexibility of the diffusion model by trying out different combinations of population specific vs. global

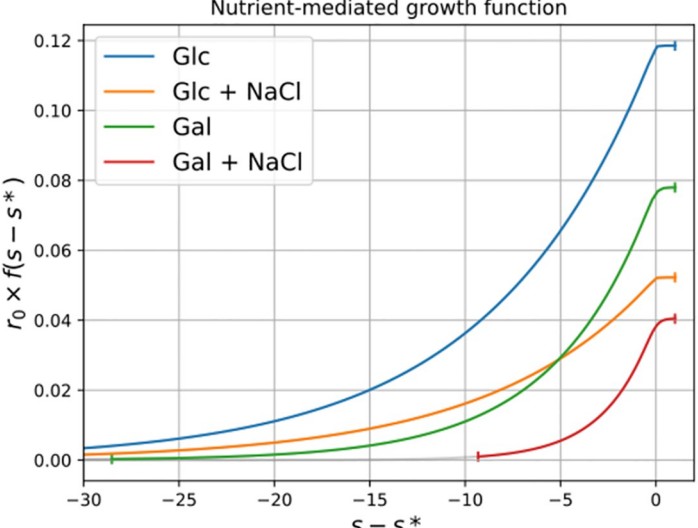

**Fig 5. Coupling functions between nutrient concentrations and population growth rates.** As part of the diffusion model we infer, for each of the four experimental environments, the coupling between resource concentration and growth (see Methods). Each curve represents this coupling function according to the parameters derived from the fit for each environment, times the $r_0$ parameter of the populations of a plate. As the nutrient and/or energy resource levels are latent and not directly observed, the units of $s$ are arbitrary and thus set to start at 1. Shift of the latent dynamics by a constant $s^*$ leaves the fits invariant (see Methods).

**Table 2. ML regressions performed to better understand drivers behind variability of $c_i$ parameter.**

| Regression model | $R^2$ | Glc | Glc + NaCl | Gal | Gal + NaCl |
|---|---|---|---|---|---|
| $x_i \mapsto c_i$ | train | 0.971 | 0.752 | 0.984 | 0.982 |
| | test | 0.834 | -46.161 | 0.934 | 0.936 |
| $N_i(t = 0) \mapsto c_i$ | train | 0.812 | 0.808 | 0.866 | 0.797 |
| | test | -0.502 | 0.101 | 0.074 | -0.212 |
| $x_i \mapsto N_i(t = 0)$ | train | 0.967 | 0.961 | 0.967 | 0.951 |
| | test | 0.737 | 0.780 | 0.797 | 0.675 |

parameters for the adjustment function $\alpha$ (Eq 1) and comparing each to our primary choice of both $r_0$, $m$ global and $c_i$ local. Keeping all three parameters local only marginally lowers the total fit error (by 7%), only $r_0$ local increases the fit error substantially (the fits do not converge properly), and only $m$ local gives essentially the same error as only $c$ local (0.04% difference). Therefore, both $c$ and $m$ could serve as the population specific parameter from the perspective of fitting. However, given the biological interpretation of $c$ denoting physiological state and $m$ the rate how that state can change, we prefer the model where only $c$ is a population specific, i.e., local, parameter and conclude that it enables us to fit the data almost perfectly. To investigate possible underlying reasons for the remaining local parameter, $c_i$, we performed a set of ML regressions. The regression models were: i) $N_i(t = 0) \rightarrow c_i$, ii) $x_i \rightarrow c_i$, iii) $x_i \rightarrow N_i(t = 0)$ which are shown in Table 2. Based on the regression analysis we conclude that both $N_i(t = 0)$ and $c_i$ have a spatial component, which we believe reflects the pre-culture protocol, but the evidence for $N_i(t = 0)$ directly driving variability in $c_i$ is not substantial.

The observation that $c_i$ has a spatial component suggests that the initial spatial feature importance seen in the ML analysis (Fig 2A) is caused by it. As our diffusion model is mechanistic it allows us to simulate data under specific scenarios (Methods). A synthetic data set, where we replaced local $c_i$ values by their plate averages, removes the initial spatial components from the ML analysis (S4 Fig). Finally, re-analysing a synthetic data set, simulated using the parameters from our real data inferred diffusion model, further demonstrates veracity of the implementation as well as self-consistency of the approach (S3 Fig).

## Discussion

Experiments conducted in a shared environment should always account for the possibility for interactions, mediated by the shared environment, to arise between the experimental units (here cell populations) they consider, as isolated as these individual units may seem. The Scan-o-matic platform is an example of a framework that intentionally creates such a shared environment in which microbial cell populations can expand as separated colonies. So far, creators of this particular and similar platforms have taken the spatial effects resulting from interactions between cell populations into consideration by using heuristic approaches, such as using control populations densely spaced along a grid [10] or by applying a posterior approach that assumes an edge effect, where only the populations at the periphery of the colony grid are affected by their position [2]. In this study, we argue that the shared environment and its effect on the growth of populations on a plate should not merely be treated as a bias induced by experimental constraints, but as a feature, which could drive spatially aware experiments.

To this end, we formulated the phenomenon of growth variation between populations as a regression problem. The field of machine learning provides methods to address this class of problems, which we utilised for demonstrating the importance of the location of a population

at different growth stages and modelling this variation. To actually explain that variation in growth, we devised a set of mathematical models, which involved a latent variable, describing the resource concentration.

Such a variable has been used to describe interactions between populations growing in a shared environment [25], and defines a class of models known as consumer-resource models [40]. These classical models can be extended to a spatial setting with a diffusion term, which allows the neighbouring units to interact via consumption and production of nutrients and energy sources [33, 41]. Chacón et al. demonstrated how the variation in colony size can be explained by such an ecological effect [33]. Here we showed how the entire spatial variation across the whole growth plate can be explained with latent resource dynamics, which included both the ecological effect of interacting neighbours as well as a boundary effect related to the physical construction and layout of the plate. This provided us an efficient, low-parametric model for understanding the determinants of spatial growth and allowed us to infer, genotype-specific, intrinsic growth parameters as well as dissect their temporal importance. Thus, variability in growth across the populations of a plate is accounted for by the interactions between neighbouring populations resulting from this diffusion process. As a population consumes resources available from its local environment, neighbouring environments and local environments diffuse—the resources consumed by the neighbouring populations become increasingly sparse in the local context. Hence, the populations interact across a plate. Due to the physical construction and layout of the plate, populations are exposed to a larger flow of nutrients and energy sources the further out in the grid they are located, which explains the variability pattern in growth across a plate.

It is important to delineate carefully between effective and intrinsic growth parameters. For instance, the growth curves analysed here can be fitted rather effectively using a generalised logistic growth model, i.e., Richards's growth model [42] which has five free parameters, and fitting it independently to each population. However, such an approach will not lead to identification of genuine intrinsic growth parameters but instead effective parameters that exhibit variability due to non-identical physiological states of the populations and differences in local resource environments. Directly modelling the environment makes the inference of intrinsic, genotype-specific, growth parameters possible and reduces the dimensionality of the model substantially. Our final model has only a single population specific parameter, the physiological state, which could possibly still be changed to a global one by improvements of the pre-culture protocol. Our study demonstrates an almost full decomposition of growth variation by analysing isogenic strains in the same plate. Other growth assays or culturing in separate plates may be needed when studying phenotypically heterogeneous strains or species to reduce variance especially if the resource usage profiles or the growth of the strains would differ greatly as then a single latent resource dimension would likely not suffice. For example, certain growth characteristics, e.g., half-saturation concentrations, can vary across orders of magnitude even for the same organism and resource [43].

The location-aware population growth model we provide relies on particular definitions of the functions $F(N_i(t))$ and $f(s(x_i, t))$. The first function, $F(N_i(t))$, describes the consumption of resources by a population and is a combination of two expressions: the nutrient consumed for population size growth [38, 44] and the nutrient consumed for population size maintenance. The second function, $f(s(x_i, t))$, couples resource availability and population growth, and was inspired by existing consumption models [45, 46]. We have tested different coupling functions for this purpose, in particular the Monod model. Unfortunately, involving the latter into a diffusion model did not produce good fits. We therefore opted for a generic form of a monotonically decreasing nonlinear function. As a caveat we note that possible simultaneous modifications to forms of $f$ and $F$ functions could in principle result in other excellent fits to

the data. Thus they are not in that sense uniquely determined and such a potential degeneracy is hard to fully avoid given that the resource variable is not directly observed. In any case, the forms we are using here are simple and demonstrate clearly that a spatial coupling mediated by diffusion and a local physiological state are able to explain spatial growth variability observed in our study.

The diffusion model, being mechanistic, has the merits of interpretability and low complexity compared to the ML approach. It can also be used to simulate growth under different scenarios. The downside is that it requires a relatively high degree of effort to arrive at such a low-complexity model, performing nonlinear regression across theoretical models as suggested in the literature, whereas ML is effortless to use in a more generic way. Symbolic regression is emerging as an automated way to discover such mechanistic models with notable success, e.g., in physics problems [47]. This seems to be a promising direction to pursue in the future and would prevent a degree of arbitrariness that is entailed by performing nonlinear regression by hand (e.g., the forms of $f$ and $F$ functions). However, we note that utilising the latent space and its dynamical equations (here Eq 7) are not yet part of standard symbolic regression toolkits, although it has already been shown that a reconstruction of latent variables is indeed possible using custom symbolic regression techniques [48, 49].

We originally aimed to optimise the usage of data from the Scan-o-matic platform by understanding details regarding the observed spatial variability. During the process, we also realised the opportunity to study the effect of spatial location for competing populations in a shared environment as genuine biological phenomenon rather than a technical problem. This opens up possibilities for spatially driven eco-evolutionary experiments on a massively parallel scale. In such experiments, each plate would correspond to a single replicate of the spatial ecosystem so one would need to measure several independent plates. In the future, one could compare outcomes of serial transfer evolution experiments with differently distributed populations across plates. One example would be invasion experiments, where one would compare the outcome of fronts of different species competing for each other's resources versus species evenly distributed across the plate. This would allow testing the extent to which intraspecific competition shapes evolution of growth strategies as quantitative traits. Another possibly interesting direction would be to place initially slow-growing but eventually high-yield strains to neighbourhoods of initially fast-growing but lower-yield ones and vice versa. This would allow addressing growth yield trade-offs in a spatial setting.

## Materials and methods

### Data sets analysed

The data sets considered in this work originate from the Scan-o-matic platform [10]. They can be retrieved by following a procedure detailed at the GitHub repository hosting our code, at https://github.com/fborse/spatial-growth/. The README document there provides a hyperlink to a repository containing multiple files, and explains which one to download.

These data sets all represent 32 x 48 grids of *S. cerevisiae* populations growing on 4 different agar plates (119mm x 82.5mm)—the grids being of size 105.75mm x 69.75mm, this leaves approximately 2.2mm of space between adjacent populations and approximately 6.5mm for the rim region. Each plate contains a different combination of utilisable energy source and presence or absence of growth-limiting salt. This results in the following combinations: 2% glucose only, 2% glucose + 1 M NaCl, 2% galactose only, and 2% galactose + 1 M NaCl. The growth medium is a synthetic growth medium, with all required nutrients other than carbon/energy being present in excess, and with the pH being buffered to 5.8. An image, based on

light transmission, of each plate is taken every 20 minutes for a total duration of 72 hours, leading to 218 data points for every population on each plate.

The four plates were generated from the same pre-culture plate. They will therefore share some growth features that vary depending on the properties of pre-culture plates (see Fig 1C). The shared features may encompass multiple colonies in a row or column wise fashion, and then typically derive from a systematic effect on initial population size or the physiological state of the populations.

The background-subtracted and calibrated population size measures are provided directly by Scan-o-matic as a 4-dimensional NumPy [50] array of 4 x 32 x 48 x 218 dimensions. This represents the 4 plates involving the 1536 growing populations set on 32 x 48 grids, ordered here as: Gal + NaCl, Glc only, Glc + NaCl, and Gal only. Population size measures are estimated from the plate images via a calibration function [10].

We further computed time derivatives (absolute growth rate) $\frac{\Delta N_i(t)}{\Delta t}$ where $\Delta t = 1$, for the population $i$ at time $t$. Additionally, we computed relative growth rates $\rho_i(t) = \frac{\Delta N_i(t)}{N_i(t)}$. All the growth curves are, unless specified otherwise, smoothened using an averaging window of 10, and discarding the 10 last data points, before further calculations are made, including the derivatives described above.

## Training the random forest regression model

The regression model chosen to predict relative growth rates $\hat{\rho}_i(t)$ comes from the scikit-learn library [51] as the RandomForestRegressor class in sklearn.ensemble. It requires only limited steps and data while still providing often very good prediction results [52, 53]. Additionally, the implementation includes access to the calculation of a set of importances for its input features [53], which are calculated by using mean decrease in impurity.

We first instance the class with its default parameters, and directly train it using its fit method on the train data. We obtain the predictions using its predict method, and the $R^2$ scores using its score method on the test data. Finally, we take advantage of the already implemented feature_importances_ property to obtain access to the importance of each input feature.

The train and test data sets are obtained by dividing the population grid into a set of contiguous adjacent non-overlapping 2 x 2 squares, and choosing for each square the bottom right position as the testing sample, while the three others become training samples. This splitting procedure ensures thus an even distribution of kinds of growth curves, as spatial effects can be assumed to be symmetrical in both directions of the grid.

## Fitting the mechanism-free models

**Fitting of $\alpha_i(t)$.** All models, except for the null model, involve a Baranyi & Roberts type intrinsic growth term defined in Eq 1. In order to obtain its $r_0$, $m$ and $c_i$ parameters, we first fit curves to a subset of the relative growth rate $\rho_i(t)$ functions, namely the initial part preceding the time of maximum local growth. We thus first perform population-specific fits to obtain locally optimal $\hat{r}_{0,i}$, $\hat{m}_i$ and $\hat{c}_i$. Then we use the latter set of parameters $\hat{c}_i$ to estimate global $r_0 \approx \hat{r}_0$ and $m \approx \hat{m}$ values. In turn, we reiterate the fits using these global parameters to estimate a new set of $c_i = \hat{c}_i$ parameters. This allows for the computation of $\alpha_i(t)$.

**An iterative method to obtain $\epsilon_k(t)$ and $v$.** We fit the density-dependent model iteratively by first fitting the $\epsilon_k(t)$ parameter and then the $v$ parameter independently. The optimal parameter values obtained from fitting the $\alpha_i(t)\epsilon_k(t)$ model are reused as initial values $\hat{\epsilon}_k(t)$ at the start of the iteration. This allows computing a first estimate $\hat{v}$ to initialise the $v$ parameter.

Subsequently, we keep this $\hat{v}$ value fixed while fitting for new parameter values $\hat{\epsilon}_k(t)$, and then keep these latter parameters fixed while fitting for a new $\hat{v}$ value. These two steps are repeated as an iterative process. At the end of that process we set the fitted values as estimates for the $\epsilon_k(t) \approx \hat{\epsilon}_k(t)$ and $v \approx \hat{v}$ parameters. An additional step performed during each iteration is a recomputation of the $\alpha_i(t)$ parameters $r_0$, $m$ and $c_i$, and their resulting $\alpha_i(t)$. This recomputation step is performed for every subsequent model—population and location-dependent model, density-dependent model and diffusion model.

## Fitting the diffusion model

Uniting the growth phenomenon under one continuum of environments—through the addition of the diffusion term $D\nabla^2 s(x_i, t)$—transforms growth models otherwise not explicitly dependent on location in the grid, into one model for each plate of explicit spatial nature. This in turn requires considerations on how to approximate the structure of a growth medium plate and its populations grid spatially.

We choose a coarse approach for the diffusion phenomenon: the grid remains a square lattice where the distances between points in the grid are assumed to all be of the same unitary length. This allows us to reformulate the diffusion term as its mean-field equivalent $D(\bar{s}(x_i, t) - s(x_i, t))$ where $\bar{s}(x_i, t)$ represents the average of the neighbours of a population $i$.

Moreover, we incorporate the lifeless space of the plate into our model by extending the grid with 3 adjacent series of empty points in all directions, resulting in a new grid of 38 x 54 dimensions, where the inner 32 x 48 points are populated in the same manner as the experimental data and the outer points do not contain colonies.

In order to fit the diffusion model, we devise a doubly iterative algorithm, where the outer iteration is set to find the optimal global parameter values for $D$, $v_1$, $v_2$, $K$ and $\kappa$, assessing the goodness of fit by comparing the predicted $\hat{\rho}_i(t)$ with the $\rho_i(t)$ rates calculated from the experimental data.

To that end, we need optimal values for $\bar{s}(x_i, t)$ and $s(x_i, t)$ for every step of such an outer iteration. The inner iteration is thus set to find these optimal values for $s(x_i, t)$ and $\bar{s}(x_i, t)$. This is achieved by starting the iteration process by setting $\bar{s}(x_i, t)$ to some initial value $s_0$ which represents the initial concentration of the nutrient, and use these values to compute $s(x_i, t)$ which allow recomputing new $\bar{s}(x_i, t)$. This latter part is then iterated until the predicted $\hat{\rho}_i(t)$ do not improve when compared to the $\rho_i(t)$ rates.

The calculation of $s(x_i, t)$ requires integration, which we perform numerically by first setting $s(x_i, 0)$ to the same $s_0$, and then applying Euler's algorithm to obtain nutrient concentrations for the rest of the time series.

**Coupling functions between nutrient concentrations and population growth rates.** In the density-dependent model, we assumed that population growth occurs at the linear part of the Monod function and we couple it to a set of mechanism-free parameters $\epsilon_k(t)$. The removal of these mechanism-free parameters causes here the need for a more elaborate coupling function $f(s(x_i, t))$, which describes the effect of nutrient availability on population growth, which we define as a generic monotonic nonlinear function:

$$f(s(x_i, t)) = \left(1 + e^{-Ks(x_i, t)}\right)^{-\kappa} \tag{10}$$

We note that one can add any constant to initial resource concentration $s(t = 0)$ provided that one subtracts it from the $s$ in the growth law function $f(s)$, so $s(t = 0) \rightarrow s(t = 0) + s^*$ and $f(s) \rightarrow f(s - s^*)$, shift the latent dynamics by a constant $s^*$ but leave the growth model for $N(t)$ exactly invariant. This means that we cannot fix this constant, i.e. unit of $s$ are arbitrary, and in the actual fits we used $s^* = 0$.

## Model-derived simulation of population growth

In order to demonstrate the self-consistency of our diffusion model fit, we used the inferred parameters (Table 1) to simulate the growth of the microbial (here yeast) populations under the diffusion model. The resulting synthetic data set reflects the experimental setup used by the Scan-o-matic platform, which consists of populations spread across an agar plate in a 32 x 48 grid. As described in the subsection describing the diffusion model, three outer layers of grid points are added to the aforementioned grid to represent the unoccupied space on the borders of a physical agar plate—these additional locations contain nutrients but no colonies. The nutrients are simulated to diffuse over time across the plate, including through and from the unoccupied locations. Therefore, the resulting grid is of size 38 x 54, to account for the border colonies having an increased amount of diffusing nutrients. Only the inner 32 x 48 locations are considered when extracting growth trajectories, as the additional space is colony-free. The diffusion process is simulated by applying the mean-field approach on the diffusion term of the model. This involves a mean-field term $\bar{s}(x_i, t)$, which describes the average nutrient concentration across the eight immediate neighbours of a location $x_i$ at a given time $t$.

For each of the four simulated plates, the initial nutrient concentration $s_0$ in every location is set to 1 and the corresponding initial population sizes $N_0$ of the inner 32 x 48 grid are taken from the original Scan-o-matic data. Unlike the method introduced in the subsection describing the diffusion model, the simulations performed here use a 10 times shorter integration time step, which corresponds to steps of 2 minutes. In order to transform the resulting simulation output to comply with our methods in terms of the unit of time, the population size trajectories are sub-sampled back to the original 20 minutes step, or 218 time points. We then used our inference methods to recover very closely the input parameters and, for completeness, the steps of the analysis outlined in the main part of this manuscript were repeated for the synthetic data (S2 Fig). Thus our inference approach is self-consistent.

## Supporting information

**S1 Table. Master table.** All error rates displayed for all 4 environments are obtained by using the SSEs between the predictions of the respective models for the values of $\hat{\rho}_i(t)$ and the $\rho_i(t)$ values calculated from the experimental data. Similarly, the main models are compared as against the null model using AIC (smaller value is better). The scores indicate that the diffusion model is the best model for all environments except Gal + NaCl where AIC would choose the density-dependent model. However, as the density-dependent model has the spatial component modelled with a "black box" of $\epsilon_k(t)$ we prefer the fully mechanistic diffusion model also for the salt-containing galactose environment.
(PDF)

**S1 Fig. Model results for all 4 plates.** Each subfigure represents, for each plate, the sum of squared errors (SSE) between the relative growth rates $\rho_i(t)$, calculated from the experimental data, and computed rates $\hat{\rho}_i(t)$, as predicted by our models, namely: 1) the null model, which consists of simply averaging for every time point the growth rates across a plate, 2) the $\alpha\epsilon$ model, which represents the $\alpha_i(t)\epsilon(t)$ model where the latter component is a mechanism-free parameter, 3) the $\alpha\epsilon_k$ model, which represents the $\alpha_i(t)\epsilon_k(t)$ model where the mechanism-free approach relaxes the global constraint of the parameter, 4) the $\alpha\epsilon_k s$ model, which adds a nutrient consumption term, 5) the diffusion model, which further removes the mechanism-free parameter and represents its effect through a diffusion process, and 6) the random forest model, which consists of a RandomForestRegressor from scikit-learn trained on 75% of the data and then predicting 25% of the growth rates by using the location on the plate and the

current population size as input features. **A—D** The spatial representation of the fitting errors, where the SSE are computed for each population individually for all the time points. Here, the predictions made by the random forest model are obtained for both the training and testing data. **E** The temporal representation of the fitting errors, where the SSE are computed for all the populations of a plate at every time point. Here, the predictions made by the random forest model are obtained for both the training and testing data.
(PDF)

**S2 Fig. Results for the synthetic dataset.** Each subfigure represents, for each plate, the sum of squared errors (SSE) between the relative growth rates $\rho_i(t)$, calculated from the data obtained by simulation, and computed rates $\hat{\rho}_i(t)$, as predicted by our models, namely: 1) the null model, which consists of simply averaging for every time point the growth rates across a plate, 2) the $\alpha\epsilon$ model, which represents the $\alpha_i(t)\epsilon(t)$ model where the latter component is a mechanism-free parameter, 3) the $\alpha\epsilon_k$ model, which represents the $\alpha_i(t)\epsilon_k(t)$ model where the mechanism-free approach relaxes the global constraint of the parameter, 4) the $\alpha\epsilon_k s$ model, which adds a nutrient consumption term, 5) the diffusion model, which further removes the mechanism-free parameter and represents its effect through a diffusion process, and 6) the random forest model, which consists of a RandomForestRegressor from scikit-learn trained on 75% of the data and then predicting 25% of the growth rates by using the location on the plate and the current population size as input features. **A—D** The spatial representation of the fitting errors, where the SSE are computed for each population individually for all the time points. Here, rhe predictions made by the random forest model are obtained for both the training and testing data. **E** The temporal representation of the fitting errors, where the SSE are computed for all the populations of a plate at every time point. Here, the predictions made by the random forest model are obtained for both the training and testing data.
(PDF)

**S3 Fig. Obtained diffusion model parameters.** The diffusion model introduced in this article uses 7 global parameters—$r_0$ and $m$ are involved in the $\alpha_i(t)$ term, $D$, $v_1$ and $v_2$ are involved in the calculation of $\partial s/\partial t$, and $K$ and $\kappa$ are involved in the $f(s)$ function—which are inferred by fitting the model on two different data sets: first the Scan-o-matic data, labelled here as experimental, and then the same synthetic data set as in S2 Fig which was simulated with the real data inferred parameters. The inferred synthetic parameters are within a reasonable range from the inferred experimental parameters thus demonstrating veracity and self-consistency of the inference approach. This is true in particular for the parameters $r_0$, $m$, $D$ and $v_1$, while the other three, $v_2$, $K$ and $\kappa$ show a degree of interplay.
(PDF)

**S4 Fig. Mean $\alpha(t)$ adjustment function parameters.** The adjustment function $\alpha(t)$ depends on three parameters $r_0$, $m$ and $c$. Throughout the article, we chose to make the former two parameters global across a plate, while the latter was made local to a population $i$. By generating synthetic population growth data according to our diffusion model parameters, but setting $c_i = \bar{c}_i$ (plate mean) for all populations of a plate, we can contrast the results from Fig 2A by using a similar ML-based approach on the generated synthetic data. **A** A synthetic data set where all 32x48 populations across a plate are simulated. While the general patterns between relative importances of population size $N_i(t)$ and location obey to the same dynamics, there is a notable difference for the initial phase. The removal the physiological state related variability, by forcing a global $\alpha(t)$ adjustment function, removes the spatial feature being important early on. **B** A similar synthetic data set, but where only every fourth population is present—other locations are left unoccupied—by subdividing the 32x48 grid into 2x2 lattices, and keeping

only the lower-right corner. The dynamics of the relative feature importances follow a similar pattern as for the full plate but as expected the spatial component, which reports diffusion, start to grow later.
(PDF)

**S5 Fig. Logarithmic representation of the growth data.** Spatial patterns of population growth on semi-solid nutrient medium. A total of 1536 isogenic populations were pinned in a 32 x 48 grid onto each of 4 plates, containing semi-solid nutrient media with small variations, and cultivated for 72 h with measurements of population size taken every 20 min. The nutrient media of the plates contain respectively 2% glucose only, 2% glucose + 1 M NaCl as growth-limiting substrate, 2% galactose only, and 2% galactose + 1 M NaCl (data from [10]). **A** Plate averages of population size estimates $N_i(t)$, absolute growth rates $\Delta N_i(t)$ and relative, per capita, growth rates $\rho_i(t)$, at each of the 218 time points. **B** Population size $N_i(t)$ for all populations on each plate, coloured by layer of equivalent distances to the nearest grid border. Darker curves are closer to the border and exhibit greater growth.
(PDF)

**S6 Fig. Comparisons between salt-free and salt-containing environments.** The growth experiments have been run on two sources of nutrient, glucose and galactose. Additionally, both nutrients have been tested in conjunction with salt or without. **A** The nutrient diffusion $D(\bar{s} - s)$ term averaged per layer as in Fig 4A. The 16 layers are for both the salt-free and salt-containing environments represented as a curve getting darker the further the layer is from the grid border. **B** The respective contributions to nutrient consumption of population growth and population maintenance, as in Fig 4B. The colours are chosen to group curves according to their consumption term. That is, blue and green for population growth, and orange and red for population maintenance.
(PDF)

**S1 Protocol. Pseudo-code for the fitting algorithms.**
(PDF)

## Acknowledgments

We would like to thank M. Manhart, D. Moradigaravand, H. Uecker, J. Vanhatalo and B. Waclaw as well as the participants of Experimental Evolution & Community Dynamics Seminar (EECD 2023, Tvärminne, Finland) for discussions. The authors wish to acknowledge CSC—IT Center for Science, Finland, for computational resources.

## Author Contributions

**Conceptualization:** Florian Borse, Ville Mustonen.

**Formal analysis:** Florian Borse, Ville Mustonen.

**Funding acquisition:** Ville Mustonen.

**Investigation:** Florian Borse, Dovydas Kičiatovas, Teemu Kuosmanen, Mabel Vidal, Guillermo Cabrera-Vives, Johannes Cairns, Jonas Warringer, Ville Mustonen.

**Methodology:** Florian Borse, Dovydas Kičiatovas, Teemu Kuosmanen, Ville Mustonen.

**Software:** Florian Borse, Dovydas Kičiatovas.

**Supervision:** Ville Mustonen.

**Writing – original draft:** Florian Borse.

**Writing – review & editing:** Dovydas Kičiatovas, Teemu Kuosmanen, Mabel Vidal, Guillermo Cabrera-Vives, Johannes Cairns, Jonas Warringer, Ville Mustonen.

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
