## [Decision Letter · Decision Letter 0]

20 Nov 2023

Dear Professor Mustonen,

Thank you very much for submitting your manuscript "Quantifying massively parallel microbial growth with spatially mediated interactions" for consideration at PLOS Computational Biology.

As with all papers reviewed by the journal, your manuscript was reviewed by members of the editorial board and by several independent reviewers. In light of the reviews (below this email), we would like to invite the resubmission of a significantly-revised version that takes into account the reviewers' comments.

We can only consider a revised version if it convincingly addresses all issues raised by the reviewers. While all reviewers appreciate this work and see some potential, it will be crucial to clarify the novelty and broader relevance of this work and point out implications for future experiments (following the specific suggestions of Reviewer #2). Please be sure to include all relevant data and computational code; several reviewers have pointed out that the original experimental data are not currently available.

We cannot make any decision about publication until we have seen the revised manuscript and your response to the reviewers' comments. Your revised manuscript is also likely to be sent to reviewers for further evaluation.

Sincerely,

Tobias Bollenbach

Academic Editor

PLOS Computational Biology

Natalia Komarova

Section Editor

PLOS Computational Biology

We can only consider a revised version if it convincingly addresses all issues raised by the reviewers. While all reviewers appreciate this work and see some potential, it will be crucial to clarify the novelty and broader relevance of this work and point out implications for future experiments (following the specific suggestions of Reviewer #2). Please be sure to include all relevant data and computational code; several reviewers have pointed out that the original experimental data are not currently available.

Reviewer's Responses to Questions

**Comments to the Authors:**

Reviewer #1: Review summary

Recent experimental advancements have enabled measuring the growth curves of microbial colonies in a massively high-throughput way. However, analyzing such big data with proper correction of noises and biases is still an open problem. This paper disentangled the spatial variability in the previously published “Scan-o-matic” growth assay using machine learning regression models, and further developed explicit mechanistic models with growth profiles and nutrient diffusion that successfully fit the data. Importantly, their computational approaches have great potential to infer various interesting quantities, such as the populational variation of lag time, the nutrient uptake rate, and the nutrient diffusivity, from massively parallel growth assays. However, unfortunately, the evaluation of these quantities was not well-conducted in the paper. Instead, they emphasized the importance of spatial resource competition, which was not a very new concept in the field of ecology. Overall, the idea and method presented in the paper are meaningful in analyzing spatial variability in big data generated under a shared environment, but extensive clarification and analysis should be required on what information can be extracted from the data with their model and how reliable it is.

Major comments.

1, The connection to the existing spatial models and the novelty of the paper should be clarified.

At the end of the first paragraph on page 4, the authors claimed “Here, our main aim is to gradually build a mechanistic model to better understand these spatial effects.” However, the final model (“diffusion model”) is not very new by itself, and claiming it as a novelty (as in the 4th paragraph of Discussion) of the paper might leave a wrong impression.

The spatial distribution of nutrients has been modeled in the various contexts of ecology. For example, section 3.2 of the book "Diffusion and ecological problems: Modern perspectives” by Okubo and Levin (Springer) summarized such models in marine ecosystems. In the microbial context, “Bacterial coexistence driven by motility and spatial competition” by Gude et al. (Nature, 2020) described a similar model to this paper. The novelty of this paper should be clarified compared with existing models.

To me, the main novelty of the paper seemed the analysis of the various parameters in massively parallel growth assays. The spatial variability in the big data should contain many interesting information that needs to be extracted by a careful computational approach. The main difficulty is that the spatial variability can be originated by various sources: systematic bias (such as temperature gradient), intrinsic stochasticity of cellular processes, and spatial nutrient competition. The method in the paper can disentangle this problem and infer quantities such as the variability of cell’s physiological state (c_i), nutrient uptake rate (\\nu_1 and \\nu_2), and nutrient diffusivity (D). The evaluation of theses parameters should be presented in the paper. The detailed comments on this point are as follows (in points 2-4).

2, Are fitted parameters realistic and meaningful?

The diffusion model has 1543 dimensions to fit and can get various quantities from the big data. These parameters should be compared with experimental counterparts. For example, is D comparable to the growth limiting nutrient of each plate (plates 1-4)? What is the distribution of c_i? Does c_i have a spatial bias or not? The evaluation of the parameters would be necessary to strengthen the impact of the paper.

3, To model alpha_i(t), why was c_i assumed to vary across the positions, and why not r_0 and m?

The intrinsic growth term alpha_i models the growth profile of a colony at position i. As alpha_i has three parameters, c_i, m, and r_0, the authors should clarify why r_0 and m were assumed to be constant across the positions. Was the variation of r_0 and m small enough compared to the variation of c_i? Alternatively, can c_i be constant, and instead, can r_0 or m be varied across positions? This point is important because understanding the variability is one of the main aims of the paper (as discussed in the last two paragraphs of page 2). Assuming no spatial variability in r_0 and m needs an additional explanation or analysis.

4, Why was f modeled by eq. 17?

The form of f (how nutrient availability impacts the growth rate) was not straightforward to me. For example, Monod growth is a widely-used model, and indeed the cited paper (reference number [40] by Birol et al.) concluded that Monod growth was more appropriate than Teissier growth in their case. Also, the modified version with exponent -\\kappa was not familiar, especially given \\kappa was very small in the synthetic data set. I would appreciate an additional explanation about the model choice.

Minor comments.

5, Configuration of the “Scan-o-matic” experiment should be clarified in Fig 1 and Method.

The graphic layout of the experiments would be helpful in Fig 1 to quickly understand the setup. Also, it might be good to clarify some experimental information in Method, rather than just citing the original paper. Especially, the area of the rim region (outside the grid) and the spatial distance between adjacent grid cites are important as the paper discussed the spatial availability of nutrients.

6, The assumption of \\lambda on page 9 should be explained more.

The explanation about “linearization” of Monod growth was confusing to me. To get \\lamda = s from \\lamda = s / (K+s), K >> s and K=1 should be assumed. I think the constant K can be absorbed into another parameter, but this should be explicitly explained, especially because s_0 = 1 is used on page 14.

Miscellaneous points.

7, When index i is first introduced at the first paragraph in page 2, it should be clarified that i labels the spatial positions. Related to this, the y labels of Fig 1a should be the average of the quantities.

8, In the caption of Fig 1, “cultivated for 48 h” should be 72 h.

9, In Fig 1b and its caption, what “inverse grid distance” means is not clear and should be explained more.

10, In the caption of Fig 1c, h of t=72h should be spaced and roman.

11, In the last paragraph on page 10, Figure S1 should be S2. S1 is used for Table and there is no Fig. S1.

Reviewer #2: I reviewed this paper together with a member of my lab, so this review represents the consensus of our perspectives.

The authors re-analyze an existing dataset of colony growth and identify the forces that drive spatial variation of lag time, growth rate, and biomass yield. First, they apply a statistical approach to explain the variation in terms of spatial position and initial population size. Second, they fit a series of mechanistic models to explain the growth curve of each colony with density-dependent and resource-dependent models of growth. They find that models that capture the decline of resource concentration, together with diffusion, can almost fully explain the spatial variation in growth for the four imaged plates in their dataset.

I appreciated the goal of this research to understand the variation present in these colony growth experiments, which is a frequent challenge for many labs. The technical implementation is solid with mostly good descriptions of the methods (but see comments below for a few places needing improvement), especially the (symbolic) code given by the authors. However, I am unsure about the scientific impact of the paper as written. Although the authors explain the growth rate variation in the specific dataset to an impressive degree, they do not offer general lessons for future experiments or about spatial variation of growth in natural environments. See below for a detailed list of items related to this as well as some more minor issues.

### Major issues

1. I felt like the paper does not provide a clear take-away message for the working theoretician or experimentalist in microbial ecology. The very last paragraph of the Discussion as some thoughts on this issue, but think the paper would be much stronger if the impact is addressed earlier and throughout the Results especially to better appreciate the significance of the steps. In particular, I still have some questions that I think the authors work can address and would be valuable for the field, but appear to be missing:

a. In light of their results, should experimentalists design these experiments differently to reduce variation, or design them in a particular way to accurately account for this information?

b. I didn't find a clear explanation of where the colony growth heterogeneity comes from in the first place. Is it due to initial variation in initial population size or nutrient concentration? Can the authors quantify the relative contributions of those two factors? For example, can the authors simulate data with fewer colonies on the same area, and test if this reduces the spatial variation?

c. What is the relevant scaling between colony distance and diffusion coefficient, between growth rate and diffusion coefficient that are necessary for strong spatial variation in growth rate? This seems valuable to know for assessing the density of colonies on a plate, and how much growth variation that will lead to.

2. The relationship between the machine learning model in the first part of the Results, and the mechanistic model in the second part of the Results, are unclear. The flowchart in Fig. 1d suggests that a dependency of the mechanistic model on machine learning model outcome, but they appear to be independent approaches to the same data. Why not just start with the mechanistic model if that provides superior insights? Was starting with the ML model just intended to be pedagogical, as many readers might intuitively start there as well? Can the authors say if both models are needed or only the mechanistic approach should be considered for similar data in the future? Deciding between statistical and mechanistic approach is a recurring question and deriving such a conclusion here would add to the impact that the article can have.

3. Looking at the statistical analysis in Fig. 2, I am lacking a null expectation as to how much variation should be explained, for example, by the initial population size and what is "suprising" in the panels of Fig. 2a,b. With the mechanistic model in the second half, the authors have the ability to create synthetic data. What happens, if they apply the same machine-learning random forest pipeline on a synthetically created dataset with logistic growth? Such a plot, potentially in the supplement or as a single panel in Fig. 2, would explicitly provide this null expectation. For the logistic dependence, the increase in variation explained by initial population size (N_0) at the end of the growth cycle seems to be expected, but the earlier rise and dip at the start of the growth cycle no.

### Minor issues

1. In Fig. 1a, second panel: the notation \\Delta N for absolute growth rate is misleading, since it suggests a difference in biomass without the units of time. Why not label it \\Delta N/\\Delta t as they define it in the text?

2. In Fig. 1a,b, plotting the population abundances on a log scale would help to better identify phases of steady exponential growth by eye. Since the data is already background subtracted, it should be sufficient to simply apply the logarithm to the existing values of N.

3. In Fig. 1b, I think it would be clearer to rename the variable "position" as "distance from the border."

4. In Fig. 1c, the colors cannot be compared across the 4 plates, because the color scale spans different ranges. Can all of them be normalized to the same range?

5. So from Fig. 1, we already know that the distance to the border is a good predictor of growth. The machine learning models starts with the raw positions to make a prediction. Can you check if the ML model "rediscovers" the distance to the border as a predictor?

6. In Fig. 2, can the authors define explicitly what "importance" means here? The only explanation in the Methods I saw is a technical reference to a scikit-learn variable but without a mathematical definition. Why do we need this weighting factor, and how would the re

---

## [Decision Letter · Decision Letter 1]

18 Mar 2024

Dear Professor Mustonen,

Thank you very much for submitting your manuscript "Quantifying massively parallel microbial growth with spatially mediated interactions" for consideration at PLOS Computational Biology. As with all papers reviewed by the journal, your manuscript was reviewed by members of the editorial board and by several independent reviewers. The reviewers appreciated the attention to an important topic. Based on the reviews, we are likely to accept this manuscript for publication, providing that you modify the manuscript according to the review recommendations.

Please address all remaining points raised by the reviewers, especially the central point raised by Reviewer #1. The various minor comments made by all reviewers should also be considered to improve the clarity of the manuscript.

Sincerely,

Tobias Bollenbach

Academic Editor

PLOS Computational Biology

Natalia Komarova

Section Editor

PLOS Computational Biology

Please address all remaining points raised by the reviewers, especially the major point raised by Reviewer #1. The various minor comments made by all reviewers should also be considered to improve the clarity of the manuscript.

Reviewer's Responses to Questions

**Comments to the Authors:**

Reviewer #1: Overall, the authors have clearly addressed the review comments and presented the extended analyses that strengthened the significance of the paper. The authors completely changed figure 4. The new figure 4 nicely shows the detailed behavior of D and F in the diffusion model. However, some concerns about f(s), which was shown in the previous figure 4, were not clearly discussed.

Major comment.

1,

I would appreciate an elaboration on the biological implication of f(s) function (equation 10). All the reviewers had comments on this (Reviewer 1 Comment 4, Reviewer 2 Comment 17, and Review 3 Comments D and K), but the authors did not provide a plausible answer.

In the response to Reviewer 1’s Comment 17, the authors stated that the main purpose of the diffusion model was to demonstrate the ability of a latent variable s, and the function could be arbitrary. In the response to Reviewer 3’s Comment K, the authors wrote that K and \\kappa (shape parameters of f(s)) were not that biologically interesting.

However, a reader would think whether the fitted f(s) can give biological information or is biologically realistic. The very small values of exponent \\kappa in Table 1 seemed weird, and I would appreciate a further discussion. In what situation was the form of f(s) derived in the original context? What does the small exponent \\kappa imply? To discuss this, it may be helpful to show the previous figure 4 (about the shape of f(s)) in the SI.

There could be multiple possible combinations of the form of f and F, as written in the sentences starting from “As a caveat”. However, a biological insight could support a specific choice of f and F. I appreciate the new figure 4, which shows the detailed behavior of the model. I would also appreciate if the authors could add a discussion about the shape of f(s).

Minor comments.

2,

I appreciate the extensive analyses about whether c, m, and r_0 should be local or global. The additional analyses showed the flexibility of the approach and strengthened the paper. However, it might be better to re-write the argument about the parameters.

It is not trivial that r_0 and m are global, because both should depend on the environment (as discussed in the original Baranyi and Roberts paper), which is likely to be spatially heterogeneous in the case of the Scan-o-matic experiments. I think that the spatial dependence of r_0 is extracted as \\epslion_k, and r_0 may be global by definition. m can be possibly local, but the environment should be spatially uniform at the early time before colonies grow and compete for nutrients, supporting the argument that m should be initially global. Just relying on the “biological interpretation” of m to argue m~global might be misleading.

(This is just a comment and not a suggestion) The additional analyses showed that r_0 was global, but m could be local. As m and c_i are both related to lag time, it seems that a simpler model with a single parameter for lag time (instead of two parameters c_i and m), such as the model by Hills and Wright (1994), may also fit the data.

3,

In Figures 4a and 4b, it might be helpful to have a plot (potentially in the SI) in which the four environments (or two, a sugar type with and without salt) can be compared in a single y-scale. It is especially important for figure 4b as the authors discussed the difference between the conditions with and without salt in the section “Explicit model of colony, environment and inter-environment interactions” (in the paragraph starting from “The comparison of both consumption”).

4,

Related to Reviewer 3 Comment G. I appreciate the descriptive names for the different plates, but the order seems confusing. They should be ordered based on the sugar type or the existence/absence of the salt. A minimal change from the current order could be swapping Glc and Gal to make “Gal + NaCl, Gal, Glc + NaCl, Glc”, but another order is possible.

Miscellaneous comments.

5,

When “mean field” is used as adjective, it may be better to write “mean-field”.

6,

The y label of Figure 4a should be <d> instead of D.</d>

Reviewer #2: I reviewed this paper together with a member of my lab, so this review represents the consensus of our perspectives.

The authors have made many valuable improvements to the paper, including a streamlined mathematical derivation of Eq. 6, testing the ML model on simulated data without spatial dependence as a sanity check, and further narrowing down the source of variation at the end of the results section. However, we recommend a few final revisions that address some remaining clarity issues:

1. We understand that the authors prefer to start their analysis with the ML model, but they motivate this in their introduction by stating that the ML model provides "insight" without saying specifically what that insight is. It seems like the main purpose of the ML model is to narrow down the features that need to be included in the mechanistic model (e.g., colony location, population size); if so, it would be helpful to state this explicitly. Or if we misunderstand, then it's even more important for the authors to better explain this.

2. We appreciate the "rule of thumb" for comparing the spatial heterogeneity across nutrients that the authors have added to the result section. Since it seems that the difference in growth rate $r_0$ is driving the variation in the product $D r_0 \\nu_1$, perhaps that can be an even simpler rule of thumb?

As a validation for their rule of thumb, the authors point to Fig. 4a. While we eventually saw that the plates with largest range on y-axis in Fig. 4a also have the highest product term $D r_0 \\nu_1$ with the data from Table 1, this seemed unnecessarily subtle. Why not add a box plot of individual colony growth rates, lag times, and yields across the four plates? This figure would be similar to Fig. 2b, but showing the actual variation in the estimated traits themselves (rather than the explained variation). This figure would allow one to compare the nutrient conditions and the variation they introduce directly in a single panel.

3. We see that the authors use the term "layer" to separate the colonies on each plate into five groups, depending on their distance to the nearest boundary. This makes sense, but needs to be explained more to the reader in the introducion when they show Fig. 1. Can the authors state the units of the "layer" variable in the caption to Fig. 1b? These units appear to be inconsistent across figures, as they range from 1-16 in Fig. 4a but 0.1-4 in Fig. 1b. It would also be helpful if the authors showed a schematic of a plate with the layer definition, for example by coloring the regions of the plate that belong to each layer (using a colony image similar to the first panel of Fig. 1d).

4. In Fig. 4a, the lines for layer 1 are too lightly colored to be easily visible, but this is the important layer for the reader to see here. Can the authors choose a color scheme with better contrast?

Reviewer #3: The authors have made significant improvements to the manuscript, addressing all of my comments. Below are two minor recommendations as a follow-up to the author's response to my original comments B and C.

Re: Comment B.

The authors have successfully strengthened the emphasis on their approach and incorporated the suggested references related to the coupling of population growth to the environment in the Discussion and Introduction sections. However, the abstract, as a concise summary of their findings, still maintains the original hierarchy of claims. To improve the abstract, I recommend that the authors also highlight the findings regarding temporal importance changes and their approach in the "Here we show..." sentence. If the authors state a good reason for keeping the current version (and the editor agrees with their rational), I can accept this decision.

Re: Comment C.

I can follow the authors response to my initial comment. However, given the central role of the three different growth phases, I strongly recommend that the authors clearly indicate them in Fig. 2a (and ideally also in Fig. 1a. In addition, since the authors agreed with our notion that the so-called exponential phase is not clearly discernible as such, it would be ideal if they could add a short comment to explain why a true exponential behavior is not apparent in the data due to overlaying transients.

**Have the authors made all data and (if applicable) computational code underlying the findings in their manuscript fully available?**

Reviewer #1: Yes

Reviewer #2: None

Reviewer #3: Yes

PLOS authors have the option to publish the peer review history of their article (what does this mean?). If published, this will include your full peer review and any attached files.

Reviewer #1: No

Reviewer #2: No

Reviewer #3: No

Figure Files:

Data Requirements:

Reproducibility:

References:

---

## [Decision Letter · Decision Letter 2]

19 Jun 2024

Dear Professor Mustonen,

We are pleased to inform you that your manuscript 'Quantifying massively parallel microbial growth with spatially mediated interactions' has been provisionally accepted for publication in PLOS Computational Biology.

Best regards,

Tobias Bollenbach

Section Editor

PLOS Computational Biology

Please still address Reviewer 2's remaining comment regarding the source of the experimental data.

Reviewer's Responses to Questions

**Comments to the Authors:**

Reviewer #1: The authors have clearly responded to the comments and improved the manuscript. I only have one comment associated with my previous comment (6). The comment (6) was not correctly displayed/written for some reason. It was intended to point out that D in the y label of Figure 4a might be better to have angle brackets as in Figure 1a.

Reviewer #2: The authors have adequately addressed all of my previously concerns. The authors have nicely included their code as a Github repository, but I just realized that the source of the experimental data isn’t totally clear in the paper — I believe it’s all from Zackrisson et al. 2016 (ref. 10) but I don’t find that stated explicitly in the introduction or data availability statement. It would be good if the authors can explicitly state that, as well as whether they are using the data in exactly the form that accompanies that paper as SI material, or if they are using data stored on another website or given to them directly from that paper’s authors (in which case they should include that form of the data as SI material to this paper). This is important if other authors want to reproduce their analysis or perform some variation on the exact same data.

**Have the authors made all data and (if applicable) computational code underlying the findings in their manuscript fully available?**

Reviewer #1: Yes

Reviewer #2: **No: **The source of the experimental data is not entirely clear

PLOS authors have the option to publish the peer review history of their article (what does this mean?). If published, this will include your full peer review and any attached files.

Reviewer #1: No

Reviewer #2: No

---

## [Editor Report · Acceptance letter]

17 Jul 2024

PCOMPBIOL-D-23-01623R2 

Quantifying massively parallel microbial growth with spatially mediated interactions

Dear Dr Mustonen,

I am pleased to inform you that your manuscript has been formally accepted for publication in PLOS Computational Biology. Your manuscript is now with our production department and you will be notified of the publication date in due course.

With kind regards,

Anita Estes
